# Thin and Deep Gaussian Processes

**Daniel Augusto de Souza**[*]
University College London

**Alexander Nikitin**
Aalto University

**ST John**
Aalto University

**Magnus Ross**
University of Manchester

**Mauricio A. Álvarez**
University of Manchester

**Marc Peter Deisenroth**
University College London

**João P. P. Gomes**
Federal University of Ceará

**Diego Mesquita**
Getulio Vargas Foundation

**César Lincoln C. Mattos**
Federal University of Ceará

## Abstract

Gaussian processes (GPs) can provide a principled approach to uncertainty quantification with easy-to-interpret kernel hyperparameters, such as the lengthscale, which controls the correlation distance of function values. However, selecting an appropriate kernel can be challenging. Deep GPs avoid manual kernel engineering by successively parameterizing kernels with GP layers, allowing them to learn low-dimensional embeddings of the inputs that explain the output data. Following the architecture of deep neural networks, the most common deep GPs warp the input space layer-by-layer but lose all the interpretability of shallow GPs. An alternative construction is to successively parameterize the lengthscale of a kernel, improving the interpretability but ultimately giving away the notion of learning lower-dimensional embeddings. Unfortunately, both methods are susceptible to particular pathologies which may hinder fitting and limit their interpretability. This work proposes a novel synthesis of both previous approaches: Thin and Deep GP (TDGP). Each TDGP layer defines locally linear transformations of the original input data maintaining the concept of latent embeddings while also retaining the interpretation of lengthscales of a kernel. Moreover, unlike the prior methods, TDGP induces non-pathological manifolds that admit learning lower-dimensional representations. We show with theoretical and experimental results that i) TDGP is, unlike previous models, tailored to specifically discover lower-dimensional manifolds in the input data, ii) TDGP behaves well when increasing the number of layers, and iii) TDGP performs well in standard benchmark datasets.

## 1 Introduction

Gaussian processes (GPs) are probabilistic models whose nonparametric nature and interpretable hyperparameters make them appealing in many applications where uncertainty quantification and data efficiency matter, such as Bayesian optimization [11], spatiotemporal modeling [6], robotics and control [5]. The key modeling choice for a GP prior is its covariance or kernel function, which determines the class of functions it represents. However, due to finite data and a fixed-form kernel function, the GP may not inter- or extrapolate as desired. Stationary kernels, such as the commonly used squared exponential kernel and the Matérn family, assume the existence of a constant characteristic lengthscale, which makes them unsuitable for modeling non-stationary data.

To construct more expressive kernels, we can consider hierarchical GP models (deep GPs or DGP). The most common deep GP construction is a functional composition of GP layers with standard

---

[*]Contact: `daniel.souza.21@ucl.ac.uk`

37th Conference on Neural Information Processing Systems (NeurIPS 2023).

stationary kernels that results in a non-stationary non-Gaussian process [3, 23]; for clarity, we will refer to this model type as compositional DGPs (CDGPs). However, CDGPs can show pathological behavior, where adding layers leads to a *loss* of representational ability [9, 7]. Alternatively, it is also possible to extend "shallow" GPs by making the kernel lengthscales a function of the input [19], resulting in the deeply non-stationary GP [DNSGP, 22]. Although this covariance function approach to DGPs does not degenerate with more layers, care must be taken to guarantee a positive semi-definite kernel matrix. Moreover, the induced space is not a proper inner-product space [18], which hinders the learning of useful manifolds.

In this paper, we address the shortcomings of previous DGP paradigms by retaining the flexibility provided by learning hierarchical lengthscale fields while also enabling manifold learning. Instead of pursuing neural networks to enhance standard GPs [1, 26], which may lead to overfitting [17], we retain the nonparametric formulation by using additional GPs to model a linear projection of each input onto a latent manifold. The key insight of our approach is that such projections are input-dependent and tailored towards more interpretable lower-dimensional manifolds with corresponding lengthscale fields. The resulting Thin and Deep GP[2] (TDGP) avoids the pitfalls of other DGP constructions while maintaining its hierarchical composition and modeling capacity beyond shallow GPs.

**Our contributions** are three-fold:

1. We propose TDGP, a new hierarchical architecture for DGPs that is highly interpretable and does not degenerate as the number of layers increases. Notably, TDGP is the only deep architecture that induces both a lengthscale field and data embeddings.
2. We prove that TDGPs and compositional DGPs are the limits of a more general DGP construction. Thus, we establish a new perspective on standard CDGPs while reaping the benefits of inducing a lengthscale field.
3. We demonstrate that TDGPs perform as well as or better than previous approaches. Our experiments also show that TDGP leans towards inducing low-dimensional embeddings.

## 2  Background

Gaussian processes (GPs) are distributions over functions and fully characterized by a mean function $m$ and a kernel (covariance function) $k$ [20]. If a function $f$ is GP distributed, we write $f \sim \mathcal{GP}(m, k)$. If not stated otherwise, we assume that the prior mean function is $0$ everywhere, i.e., $m(\cdot) \equiv 0$. Typically, the kernel possesses a few interpretable hyperparameters, such as lengthscales or signal variances, estimated by the standard maximization of the marginal likelihood [20].

The squared exponential (SE) kernel is arguably the most commonly used covariance function in the GP literature and, in its most general form [25], it can be written as

$$k_{\mathrm{SE}}(\boldsymbol{a}, \boldsymbol{b}) = \sigma^2 \exp\left[-\tfrac{1}{2}(\boldsymbol{a} - \boldsymbol{b})^{\mathsf{T}} \boldsymbol{\Delta}^{-1}(\boldsymbol{a} - \boldsymbol{b})\right], \tag{1}$$

where $\boldsymbol{a}, \boldsymbol{b} \in \mathbb{R}^D$, the constant $\sigma^2 > 0$ defines the signal variance, and $\boldsymbol{\Delta} \in \mathbb{R}^{D \times D}$ is a lengthscale matrix. Notably, the SE kernel is *stationary*, i.e., $k_{\mathrm{SE}}(\boldsymbol{a} + \boldsymbol{c}, \boldsymbol{b} + \boldsymbol{c}) = k_{\mathrm{SE}}(\boldsymbol{a}, \boldsymbol{b})$ for any $\boldsymbol{c} \in \mathbb{R}^D$. Furthermore, when the lengthscale matrix $\boldsymbol{\Delta} = \lambda^2 \boldsymbol{I}$, the kernel is *isotropic*, meaning that it can be written as a function $\pi_{\mathrm{SE}}(d^2)$ of the squared distance $d^2 = \|\boldsymbol{a} - \boldsymbol{b}\|_2^2$. For more details on the significance of the lengthscale parameter, we refer the reader to Appendix A.

Stationary kernels enforce invariances, which may not always be desirable. However, stationary kernels can be used as building blocks to derive broader families of kernels (including *non-stationary* kernels), either by composing them with deformation functions or through mixtures of lengthscales.

**Deformation kernels** result from applying a deformation function $\boldsymbol{\tau} : \mathbb{R}^D \rightarrow \mathbb{R}^Q$ to $\boldsymbol{a}$ and $\boldsymbol{b}$ before feeding them to a stationary kernel $k$ in $\mathbb{R}^Q$. Thus, a deformation kernel $k_\tau$ follows

$$k_\tau(\boldsymbol{a}, \boldsymbol{b}) = k(\boldsymbol{\tau}(\boldsymbol{a}), \boldsymbol{\tau}(\boldsymbol{b})). \tag{2}$$

For a linear transformation $\boldsymbol{\tau}(\boldsymbol{x}) = \boldsymbol{W}\boldsymbol{x}$, we can interpret $k_\tau$ as a stationary kernel with lengthscale matrix $\boldsymbol{\Delta} = [\boldsymbol{W}^{\mathsf{T}}\boldsymbol{W}]^{-1}$. However, for more intricate $\boldsymbol{\tau}(\cdot)$, interpreting or analyzing $k_\tau$ can be very challenging. For instance, this is the case for deep kernel learning [DKL, 26] models, in which $\boldsymbol{\tau}(\cdot)$ is an arbitrary neural network.

---

[2]"Thin" refers to the graph-theoretical girth of the graphical model of our proposed DGP construction.

**Compositional deep GPs** [CDGPs, 3] also rely on deformation kernels, but they use a GP prior to model $\boldsymbol{\tau}(\cdot)$. The kernel of the $\boldsymbol{\tau}(\cdot)$ process can also (recursively) be considered to be a deformation kernel, thereby extending DGPs to arbitrary depths. However, stacking GP layers reduces the interpretability of DGPs, and their non-injective nature makes them susceptible to pathologies [9].

**Lengthscale mixture kernels** [19] are a generalization of the process of convolving stationary kernels with different lengthscales from Higdon et al. [14]. For arbitrary isotropic kernels $k$, i.e., $k(\boldsymbol{a}, \boldsymbol{b}) = \pi_k(\|\boldsymbol{a} - \boldsymbol{b}\|_2^2)$, Paciorek and Schervish [19] construct a non-stationary $k_{\text{lmx}}$ as a function of a field of lengthscale matrices $\boldsymbol{\Delta} : \mathbb{R}^D \to \mathbb{R}^{D \times D}$, such that

$$k_{\text{lmx}}(\boldsymbol{a}, \boldsymbol{b}) = |\boldsymbol{\Delta}(\boldsymbol{a})|^{\frac{1}{4}} |\boldsymbol{\Delta}(\boldsymbol{b})|^{\frac{1}{4}} |(\boldsymbol{\Delta}(\boldsymbol{a}) + \boldsymbol{\Delta}(\boldsymbol{b}))/2|^{-\frac{1}{2}} \pi_k(\delta), \tag{3}$$

where $\delta = 2(\boldsymbol{a} - \boldsymbol{b})^\intercal (\boldsymbol{\Delta}(\boldsymbol{a}) + \boldsymbol{\Delta}(\boldsymbol{b}))^{-1} (\boldsymbol{a} - \boldsymbol{b})$. The explicit lengthscale field $\boldsymbol{\Delta}$ makes this model more interpretable than general deformation kernels. However, Paciorek [18] notes that $\delta(\boldsymbol{a}, \boldsymbol{b})$ may violate the triangle inequality and therefore does not induce a manifold over the input space. This departure from the properties of stationary kernels is due to the matrix inside the quadratic form being a function of both $\boldsymbol{a}$ and $\boldsymbol{b}$. Another caveat is that the scale of $k_{\text{lmx}}$ is also controlled by a pre-factor term that depends on $\boldsymbol{\Delta}$, allowing for unintended effects (such as unwanted long-range correlations), especially in rapidly varying lengthscale fields.

**Deeply non-stationary GPs** [DNSGPs, 7, 22] use the lengthscale mixture kernel and parameterize the function $\boldsymbol{\Delta}(\boldsymbol{x})$ with a warped GP prior to obtain a deep GP model. Similar to CDGPs, this model can be extended in depth [22, 7] by considering the kernel of $\boldsymbol{\Delta}(\boldsymbol{x})$ to be non-stationary with its lengthscales stemming from another GP. A practical issue in these models is guaranteeing that $\boldsymbol{\Delta}(\boldsymbol{x})$ is positive semi-definite. Therefore, in practice, $\boldsymbol{\Delta}(\boldsymbol{x})$ is usually restricted to be diagonal.

## 3 Thin and Deep GPs (TDGPs)

As discussed in the previous section, deep GP constructions follow naturally from hierarchical extensions of a base kernel. Therefore, we arrange the presentation of TDGPs in three parts. First, we propose a *kernel* that admits interpretations both in terms of its lengthscale and of its induced manifold. Second, we use this kernel to build a novel type of deep GP *model* (TDGP). Third, we describe how to carry out *inference* for TDGPs. Finally, we discuss *limitations* of our approach.

**Kernel.** We address the drawbacks of the approaches based on deformation and lengthscale mixture kernels — i.e., the lack of interpretability and failure to induce a manifold, respectively — by proposing a synthesis of both methods, retaining their positives whilst mitigating some of their issues. Starting from the discussion in Section 2, a squared exponential kernel with lengthscale matrix $[\boldsymbol{W}^\intercal \boldsymbol{W}]^{-1}$ corresponds to a deformation of an isotropic kernel:

$$\begin{aligned} k_{\text{SE}}(\boldsymbol{a}, \boldsymbol{b}) &= \pi_{\text{SE}}((\boldsymbol{a} - \boldsymbol{b})^\intercal \boldsymbol{W}^\intercal \boldsymbol{W} (\boldsymbol{a} - \boldsymbol{b})) \\ &= \pi_{\text{SE}}((\boldsymbol{W}\boldsymbol{a} - \boldsymbol{W}\boldsymbol{b})^\intercal (\boldsymbol{W}\boldsymbol{a} - \boldsymbol{W}\boldsymbol{b})) \\ &= \pi_{\text{SE}}(\|\boldsymbol{W}\boldsymbol{a} - \boldsymbol{W}\boldsymbol{b}\|_2^2). \end{aligned} \tag{4}$$

Therefore, this is the same as applying a linear deformation $\boldsymbol{\tau}(\boldsymbol{x}) = \boldsymbol{W}\boldsymbol{x}$.

We propose extending this to a non-linear transformation that is locally linear by letting $\boldsymbol{W}$ vary as a function of that input, so $\boldsymbol{\tau}(\boldsymbol{x}) = \boldsymbol{W}(\boldsymbol{x})\boldsymbol{x}$. This results in the TDGP kernel:

$$\begin{aligned} k_{\text{TDGP}}(\boldsymbol{a}, \boldsymbol{b}) &= \pi_{\text{SE}}(\|\boldsymbol{W}(\boldsymbol{a})\boldsymbol{a} - \boldsymbol{W}(\boldsymbol{b})\boldsymbol{b}\|_2^2) \\ &= \pi_{\text{SE}}((\boldsymbol{W}(\boldsymbol{a})\boldsymbol{a} - \boldsymbol{W}(\boldsymbol{b})\boldsymbol{b})^\intercal (\boldsymbol{W}(\boldsymbol{a})\boldsymbol{a} - \boldsymbol{W}(\boldsymbol{b})\boldsymbol{b})). \end{aligned} \tag{5}$$

Equation (5) cannot be written as a Mahalanobis distance like in Eq. (4), but in the neighborhood of $\boldsymbol{x}$, $[\boldsymbol{W}^\intercal(\boldsymbol{x})\boldsymbol{W}(\boldsymbol{x})]^{-1}$ is a lengthscale matrix, thus allowing $k_{\text{TDGP}}$ to be implicitly parametrized by a lengthscale field just like the lengthscale mixture kernels of Paciorek and Schervish [19]. Hence, it allows for better interpretability than the deformation kernels considered in compositional DGPs.

However, unlike the lengthscale mixture approach of Eq. (3), our kernel in Eq. (5) does not introduce an input-dependent pre-factor, thereby avoiding pathologies when the lengthscale varies rapidly. Moreover, since the distance induced by the quadratic form obeys the triangle inequality, it induces a manifold in the input space. Hence, it addresses the two issues of lengthscale mixture kernels.

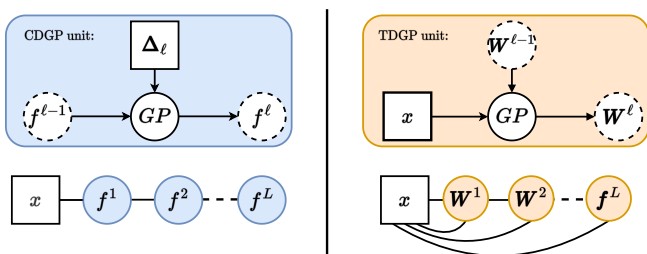

Figure 1: Graphical models for a CDGP (left) and TDGP (right) with $L$ layers. The colored boxes represent the architecture of a single layer; dashed nodes represent where layers connect. CDGP directly composes functions, whereas TDGP builds a hierarchy of input-dependent lengthscale fields.

**Model.** In an analogous manner to the compositional DGP and DNSGP, we present how to use this kernel to build a hierarchical GP model. Our $L$-layers deep model is described as follows:

$$p(f^L(\cdot) \mid \boldsymbol{h}^{L-1}(\cdot)) = \mathcal{GP}(f \mid 0, \pi_k(\|\boldsymbol{h}^{L-1}(\boldsymbol{a}) - \boldsymbol{h}^{L-1}(\boldsymbol{b})\|)), \qquad (6)$$

where

$$\boldsymbol{h}^\ell(\boldsymbol{x}) = \boldsymbol{W}^\ell(\boldsymbol{h}^{\ell-1}(\boldsymbol{x}))\boldsymbol{x}, \qquad \boldsymbol{h}^0(\boldsymbol{x}) = \boldsymbol{x}, \qquad (7)$$

$$p(\boldsymbol{W}^\ell(\cdot)) = \prod_{d=1}^{D} \prod_{q=1}^{Q} \mathcal{GP}(w_{qd}^\ell \mid \mu_w^\ell, k_w^\ell). \qquad (8)$$

Therefore, this model can be extended arbitrarily by deforming the kernel of the entries of $\boldsymbol{W}^\ell$ with another locally linear transformation with its own matrix $\boldsymbol{W}^{\ell-1}$.

Figure 1 compares the graphical models of TDGP and compositional DGP. We notice that TDGP learns hierarchical input-dependent lengthscale fields, instead of a straight composition of processes. We call this model **Thin and Deep GP** (TDGP) by the fact that our graphical model always has cycles of bounded length due to the connection of every hidden layer with the inputs $\boldsymbol{X}$. Therefore, it has finite girth, in contrast to the unbounded girth of the compositional DGP graphical model. Importantly, however, TDGP are related to CDGPs as both can be seen as locally affine deformations of the input space (proof in Appendix B).

**Theorem 3.1** (**Relationship between TDGP and CDGP**). *Any $L$-layer CDGP prior over a function $f(\boldsymbol{x}) = h^L(\boldsymbol{h}^{L-1}(\cdots \boldsymbol{h}^1(\boldsymbol{x}) \cdots))$ is a special case of a generalized TDGP prior with equal depth defined over the augmented input space $\tilde{\boldsymbol{x}} = [\boldsymbol{x}, 1]^\mathsf{T}$. Since linear deformations $\tilde{\boldsymbol{W}}\tilde{\boldsymbol{x}}$ in the augmented space correspond to affine transformations $\boldsymbol{W}\boldsymbol{x} + \boldsymbol{d}$ in the original space, the special case of the CDGP model corresponds to a TDGP where the prior variance of $\boldsymbol{W}^\ell$ approaches zero.*

Unlike DNSGP, which makes the positive semi-definite matrix $\boldsymbol{\Delta}$ a function and requires the use of warped GP priors, our hidden layers $\boldsymbol{W}$ are arbitrary matrices that admit a regular Gaussian prior. By placing zero-mean priors on the entries of $\boldsymbol{W}$, we encourage the MLE estimate to maximally reduce the latent dimensionality of $\boldsymbol{h}$. This is because the latent dimensionality becomes linked to the number of rows with non-zero variance in the prior, e.g., if the prior kernel variance $i$-th row of $\boldsymbol{W}^\ell$ tends to zero, the posterior over that row is highly concentrated at zero, eliminating the $i$-th latent dimension. In a compositional DGP, this inductive bias also corresponds to a zero-mean prior in the hidden layers. However, as discussed by Duvenaud et al. [9], and reproduced in Fig. 2, this choice introduces a pathology that makes the derivatives of the process almost zero.

**Inference.** To estimate the posterior distributions and the hyperparameters of our model, we perform variational inference (VI). We introduce inducing points $\boldsymbol{u}$ for the last-layer GP $f(\cdot)$ and inducing points $\boldsymbol{V}$ for the $W(\cdot)$ processes. Similar to how VI methods for DGPs [3] were based on the non-deep GPLVM model [24], our method builds upon VI for the hyperparameters of square exponential kernels in shallow GPs as discussed by Damianou et al. [4]. However, we replace the Gaussian prior $p(\boldsymbol{W})$ with a GP prior $p(\boldsymbol{W}(\cdot))$. For instance, the variational distribution for a two-layer TDGP is:

$$q(\boldsymbol{f}, \boldsymbol{W}, \boldsymbol{u}, \boldsymbol{V}) = p(\boldsymbol{f} \mid \boldsymbol{u}) \, \mathrm{N}(\boldsymbol{u} \mid \check{\boldsymbol{\mu}}_u, \check{\boldsymbol{\Sigma}}_u) \prod_{d=1}^{D} \prod_{q=1}^{Q} p(\boldsymbol{w}_{qd} \mid \boldsymbol{v}_{qd}) \, \mathrm{N}(\boldsymbol{v}_{qd} \mid \check{\boldsymbol{\mu}}_{v_{qd}}, \check{\boldsymbol{\Sigma}}_{v_{qd}}), \quad (9)$$

where for the final layer the parameters $\breve{\boldsymbol{\mu}}_u$, $\breve{\boldsymbol{\Sigma}}_u$ are not estimated but replaced with their closed-form optimal solutions. Appendix C contains a derivation of the ELBO and more details.

**Limitations.** For large numbers of layers, our VI scheme for TDGP may become computationally intensive. By stacking additional hidden layers $\boldsymbol{W}(\boldsymbol{x})$, we add $D \times Q$ GPs into the model; consequently, the number of variational parameters increases, which can slow down optimization. More specifically, inference uses $\mathcal{O}(L \times D \times Q \times m^2)$ parameters and takes time $\mathcal{O}(L \times D \times Q \times (m^2 + m^2 n))$ to compute the ELBO. Additional details on runtime as a function of data size and width can be found in the Appendix E. Furthermore, without the addition of a bias to the input data $\boldsymbol{x}$, TDGP performs locally linear transformations $\boldsymbol{W}(\boldsymbol{x})\boldsymbol{x}$, meaning that the neighborhood of $\boldsymbol{x} = \boldsymbol{0}$ is kept unchanged for each layer.

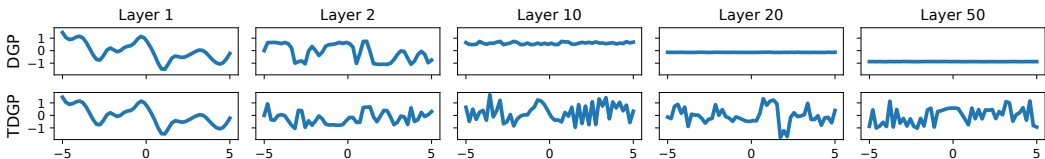

Figure 2: Samples from a CDGP prior (top row) and a TDGP prior (bottom row), both with zero mean. Each column represents the number of layers, where one layer corresponds to a regular shallow GP. When the number of layers increases, CDGP samples quickly become flat, i.e. the prior distribution no longer supports 'interesting' functions. Notably, this pathology does not occur with TDGP.

## 4   Experiments

To assess whether TDGP accurately captures non-smooth behavior and has competitive results with other GP models, we performed a regression experiment on a synthetic dataset with input-dependent linear projections and another experiment in varied empirical datasets. In this section, we examine the two-layer TDGP against a two-layer CDGP, two-layer DNSGP, DKL, and the shallow sparse GP (SGP). These models are evaluated on average negative log-predictive density (NLPD) and mean relative absolute error (MRAE). All metrics are "the lower, the better". Importantly, throughout the experiments, we also emphasize (i) the interpretability of our model compared to the prior art and (ii) TDGP's inductive bias towards learning low-dimensional embeddings. In all experiments, inputs and targets are normalized so that the training set has zero mean and unit variance. Appendix D contains more details of architecture, training, and initialization. We implemented the experiments in Python using GPflow [10], GPflux [8], and Keras [2]. Code is available as supplementary material at `https://github.com/spectraldani/thindeepgps`.

### 4.1   Synthetic experiment

**Data.** To assess our intuition that TDGP leans towards inducing low-dimensional manifolds, we show how well TDGP and competitors can fit a composite function $f = g \circ h$, where $g : \mathbb{R} \to \mathbb{R}$ and $h : \mathbb{R}^2 \to \mathbb{R}$ are non-linear functions. In this context, $h$ acts as a *"funnel"* inducing a 1D manifold. For more details on $f$ and how we sample from the input space, we refer the reader to Appendix D.

**Results.** Figure 3 shows the posterior mean of TDGP and competing methods for our synthetic dataset. Note TDGP accurately fits the target function (leftmost panel), while the other methods fail to capture the shape of several of the local maxima. Consequently, Table 1 shows that TDGP performs significantly better in terms of both NLPD and MRAE in the test data.

Table 1: Results for the synthetic data. TDGP significantly outperforms baselines.

|  | NLPD | MRAE |
|---|---|---|
| SGP | −1.49 | 0.11 |
| DKL | 36.82 | 0.21 |
| CDGP | −1.28 | 0.12 |
| DNSGP | −1.46 | 0.12 |
| **TDGP** | **−3.57** | **0.00** |

**Examining the latent space.** TDGP outperformed competitors in predictive performance, but how do their latent spaces compare? To further understand the nuances that distinguish each model, Fig. 4 plots their respective latent spaces and lengthscale fields — along with the true latent space induced by $h$ as a baseline. While TDGP induces both a latent space and a lengthscale field, it is important to highlight that the same does not hold for CDGP and DNSGP. Thus, Fig. 4 does not show a latent space for DNSGP or a lengthscale field for CDGP. Notably, TDGP's latent space perfectly captures

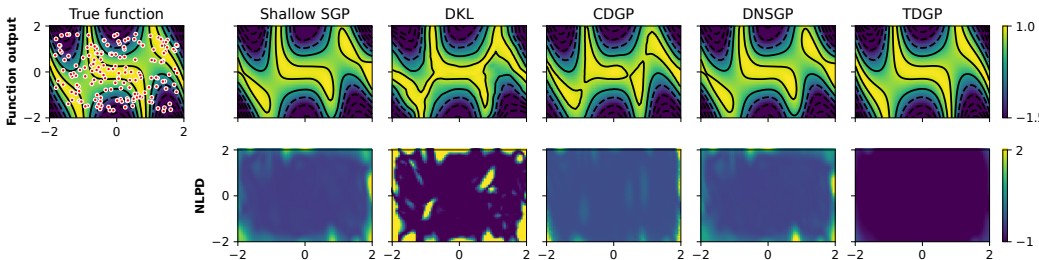

Figure 3: Synthetic 2D dataset with one latent dimension. The leftmost plot shows the true function and the location of the training data as red dots. The remaining plots show the mean predictions and NLPD of each method. TDGP presents the best fit for the true function, as observed by the mean and NLPD plots.

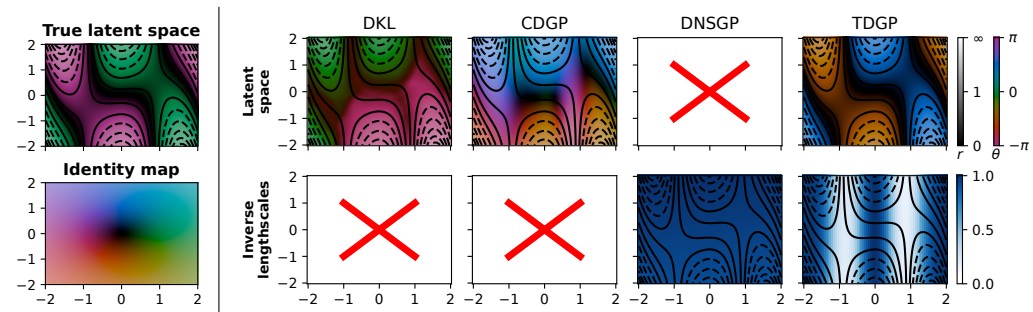

Figure 4: Synthetic dataset: true latent space (left), latent spaces learned by each model (top-right), and their inverse lengthscale fields (bottom-right). Models that do not induce a latent space or a lengthscale field are marked in **red crosses**. Note that TDGP is the only model that allows both interpretations. Furthermore, TDGP's latent space perfectly captures the shape of the ground truth.

the shape of $h$, while DKL and CDGP fail to do so. Analyzing the lengthscale fields, we conclude that TDGP successfully learns a non-stationary kernel, unlike DNSGP.

Finally, Fig. 5 shows that TDGP learns to give much higher importance to one of its latent dimensions, supporting our intuition that TDGPs lean towards learning low-dimensional manifolds. While the shallow GP and CDGP also weigh one dimension higher than the other, this discrepancy is less accentuated. It is worth mentioning that measuring the relevance of latent dimensions depends on the architecture we evaluate. For DGPs and DKL, we can use the inverses of the output layer's kernel lengthscales. Naturally, the same rationale applies to shallow GPs. For TDGP, the analogous variable is the kernel variance of each hidden layer's row, since the

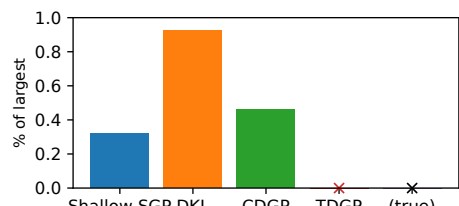

Figure 5: Synthetic dataset: what is the *effective* dimensionality of the inferred latent space? A dimension's relevance is given by its inverse lengthscale; we plot the relative relevance of the least relevant compared to the largest. Only TDGP matches the one-dimensionality of the true latent space.

larger this value is, the more distant from zero the values of that row are. Overall, these results suggest that TDGP leans towards learning sparser representations than its competitors.

## 4.2 Bathymetry case study

**Data.** As a case-study, we also apply TDGP to the bathymetry dataset GEBCO. This dataset contains a global terrain model (elevation data) for ocean and land. As an example of a non-stationary task, we selected an especially challenging subset of the data covering the Andes mountain range, ocean, and land (see Appendix D.2 for details). We subsample 1,000 points from this region and compare the methods via five-fold cross-validation.

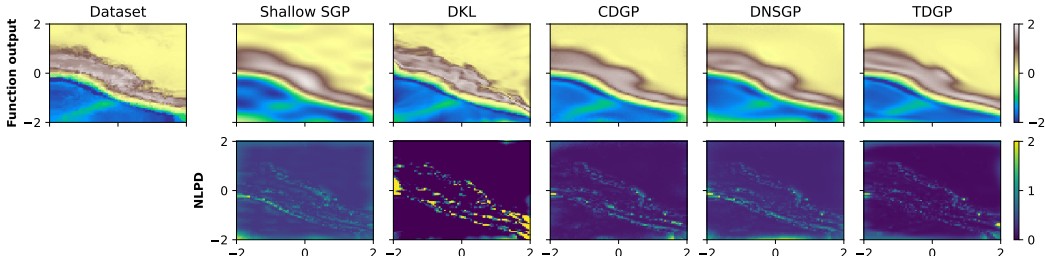

Figure 6: Results for GEBCO bathymetry dataset. The leftmost plot shows the dataset, while the remaining plots show the mean predictions and NLPD of each method. Note how, despite having the best mean posteriors, DKL has an overfitting problem, as shown in the NLPD. TDGP is the most balanced, especially when capturing the transitions between ocean/mountains/plains.

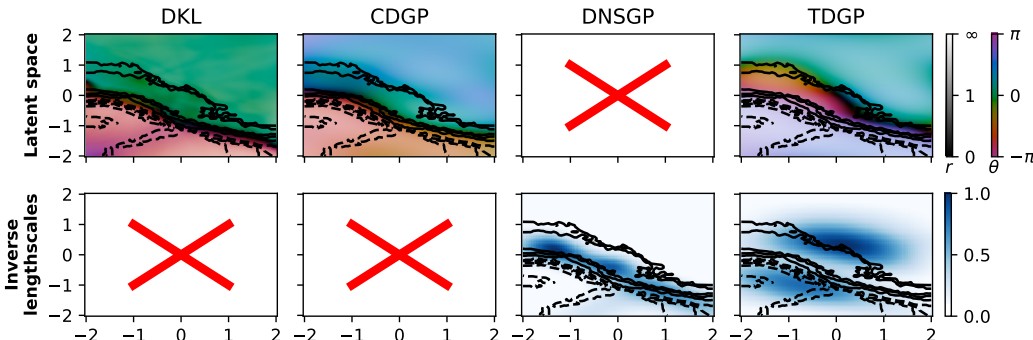

Figure 7: GEBCO dataset: visualization of the latent spaces learned by each model (top) and their inverse lengthscale fields (bottom). Models that do not induce a latent space or a lengthscale field are marked in **red crosses**. TDGP is the only model that allows both interpretations. For expert users, the lengthscale field is the only one on which informative priors can be put.

**Results.** The NLPD and MRAE results are listed in Table 2, where we observed our method to be slightly better than others. However, more importantly, this dataset can be thoroughly examined and interpreted. Figure 8 shows correlation plots for a point located on the lower slope of the Andes. We observe more sensible correlations for TDGP and CDGP compared to other methods — the correlation is high in the locations across the slope. This plot also highlights a recurring problem with DNSGP: Despite a high inverse lengthscale barrier in the Andes, there is still a correlation between the slope of the mountain and the sea level. Additionally, Fig. 7 shows the domain coloring of learned latent spaces and the sum of the eigenvalues of the lengthscale fields. Once again, we note that only TDGP can be analyzed in both ways. This is an advantage in settings where expert priors on the smoothness exist, as these can be placed directly on the lengthscale field instead of the less accessible latent space mapping. As expected, methods that learn the lengthscale field place high inverse lengthscale values around the mountain range and low values in the smooth oceans and plains.

Table 2: Performance of each model on the GEBCO dataset (avg±std). Lower is better.

|      | NLPD | MRAE |
| --- | --- | --- |
| SGP | $-0.13 \pm 0.09$ | $1.19 \pm 0.63$ |
| DKL | $3.85 \pm 0.92$ | $0.59 \pm 0.31$ |
| CDGP | $-0.44 \pm 0.12$ | $0.83 \pm 0.56$ |
| DNSGP | $-0.31 \pm 0.12$ | $1.12 \pm 0.75$ |
| TDGP | $\mathbf{-0.53 \pm 0.10}$ | $0.66 \pm 0.43$ |

## 4.3 Benchmark datasets

**Data.** We also compare the methods on four well-known regression datasets from the UCI repository. To assess each model fairly, we adopt a ten-fold separation of the datasets into training and testing.

**Results.** Figure 9 shows the average NLPD for each method in the UCI datasets along with their respective standard deviations. TDGP performs either on par with other methods or outperforms them. Figure 10 also shows the relative relevance of each method's latent dimensions. Similarly

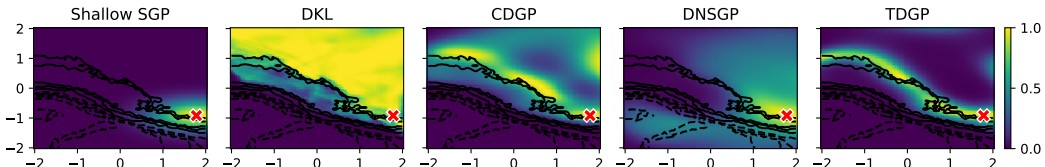

Figure 8: Correlation plots on GEBCO dataset for a datapoint marked ✖ $(1.7, -0.9)$, which is located close to mountains. We observe better interpretability with TDGP (the zone of high correlation coincides with mountains) compared to DNSGP (high correlation extends into the sea and plains).

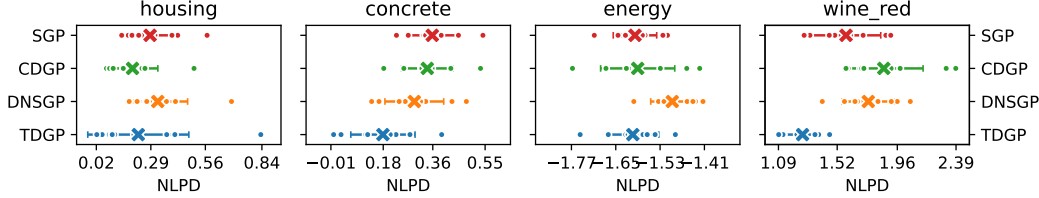

Figure 9: Benchmark datasets: test NLPD (lower is better) of each model for 10 folds. Each dot represents the result of a fold. The cross and bar represent the mean and standard deviation across all folds. TDGP is as good or better than the compared models.

to what we observed in the synthetic experiments, TDGP's inductive bias leads to a sharp split between relevant and irrelevant dimensions compared to the other approaches. Even for the cases with similar generalization errors, TDGP shows better interpretability and bias towards learning lower-dimensional representations.

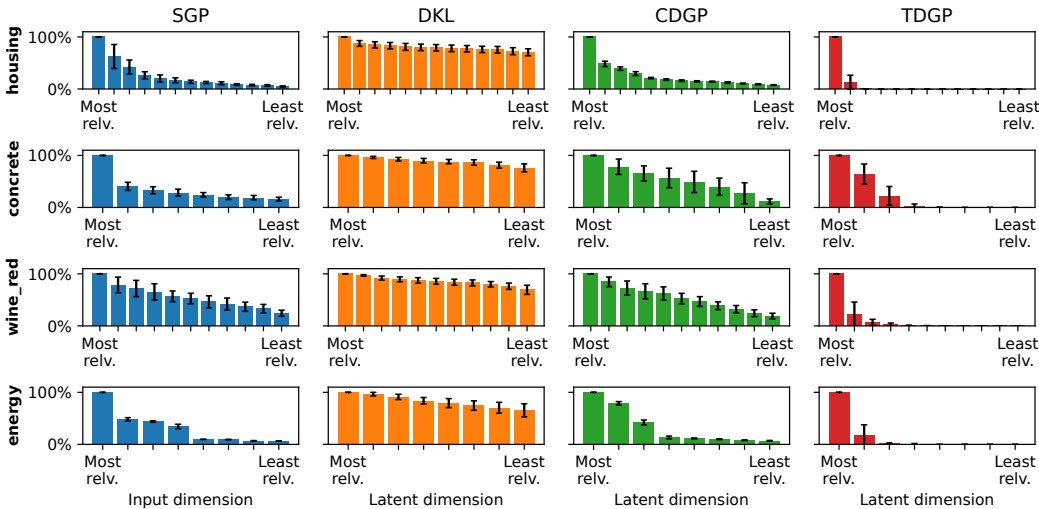

Figure 10: Benchmark datasets: comparison of the relevances of the latent dimensions identified by each model (mean and $1\sigma$ interval). TDGP consistently identifies low-dimensional latent spaces (most dimensions are irrelevant).

## 5 Related work

Expressive non-stationary GPs can be constructed in a variety of ways. We give a brief overview of some contributions available in the literature below.

**Convolution of SE and scale mixture kernels.** Non-stationary covariance learning can be done by convolving two SE kernels with different lengthscales. This well-studied approach was later extended to arbitrary kernels by Paciorek and Schervish [19]. Gibbs [12] introduced this formulation,

representing the lengthscale as the sum of squares of an arbitrary function applied to each covariance input. Later, Higdon et al. [14] introduced this approach in the spatial context, varying the focus points of the ellipse representing the $2 \times 2$ lengthscale matrix of a SE covariance with independent GPs, which constitutes a two-layer DGP in 2D. Heinonen et al. [13] introduced a method to jointly vary the covariance amplitude and lengthscale, and the observation noise over the domain, using Hamiltonian Monte Carlo for inference. As discussed in Section 2, all these methods inherit the limitations of their kernel, i.e. issues with the pre-factor and violation of the triangle inequality, which hinders manifold learning.

**Compositional deep GPs.** Compositional DGPs [15, 3] construct priors over complex functions by repeatedly applying a sequence of non-parametric transformations to the input, with each transformation being represented by a GP with a stationary covariance. One of the the most significant issues with CDGPs is that, unlike neural networks which compose linear functions through non-linear scalar activations, every latent layer is a non-linear function. Therefore, several authors have suggested improvements to DGPs by adding skip connections that directly concatenate the hidden layer with the input space [9] or the use of linear mean functions [16, 23] for improved performance and capability. CDGP research also explores improvements in inference, such as the use of mini-batching and more efficient approximations to intractable integrals [23], and the inclusion of auxiliary latent variables to allow for more flexible posteriors [21]. Since TDGP is based on deformation kernels and is related to CDGPs, we believe it can leverage many of these advances in inference, which is left to future investigations.

**Deep kernel learning.** An alternative to CDGPs is to use layers of parametric transformations to learn arbitrarily complex kernels. Deep kernel learning [1, 26] parameterizes such transformations with a deterministic neural network. Since these deep kernels have many parameters and are not constructed in a Bayesian way, they are prone to overfitting and miscalibrated uncertainties, and can perform poorly in practice [17]. Nevertheless, by using Bayesian injective warping functions that are well-suited for spatial applications, Zammit-Mangion et al. [27] were able to achieve good results in low-dimensional datasets.

# 6   Conclusion

This work presented Thin and Deep GP (TDGP), a new hierarchical architecture for DGPs. TDGP can recover non-stationary functions through a locally linear deformation of stationary kernels. Importantly, this construction allows us to interpret TDGP in terms of the latent embeddings and the lengthscale field it induces. Additionally, while TDGP shares a connection with CDGP, our prior does not concentrate on "flat" samples when the number of layers increases — as is the case with CDGP. Furthermore, our experiments show that TDGP performs as well as or better than previous approaches. Moreover, TDGP has a robust inductive bias toward learning low-dimensional embeddings, which is instrumental for better interpretability.

That being said, we expect TDGP will be especially useful for geospatial modeling in cases where we are modeling non-stationary functions and when we have expert knowledge on how this function should vary over space/time, which could be inserted as priors over the locally linear transformations. We also believe that recent improvements in inference available in the literature could greatly enhance the expressivity and ease of training of TDGP.

# Acknowledgments

This work was supported in part by the CONFAP-CNPq-THE UK Academies program (grant UKA-00160-00003.01.00/19). Diego Mesquita was supported by the Silicon Valley Community Foundation (SVCF) through the Ripple impact fund, the Fundação de Amparo à Pesquisa do Estado do Rio de Janeiro (FAPERJ) through the *Jovem Cientista do Nosso Estado* program, and the Fundação de Amparo à Pesquisa do Estado de São Paulo (FAPESP) through the grant 2023/00815-6.

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
