# OpenReview forum: "Thin and deep Gaussian processes"
_NeurIPS.cc/2023/Conference — NeurIPS 2023 poster_

### Official Review · Reviewer_DvNb · 2023-07-02

**Soundness:** 3 good
**Presentation:** 4 excellent
**Contribution:** 3 good
**Rating:** 7
**Confidence:** 4

**Summary:**

Techniques such as Deep Kernel Learning and extensions such as Deep GPs have been proposed in order to overcome the flexibility constraints associated with standard shallow GPs. However, although these techniques are better suited to model non-stationary data, they are still susceptible to issues such as pathologies when extended to more than a few layers, as well as overfitting. These techniques also lack the interpretability of the latent space and associated length-scales that make standard GPs so appealing. In order to mitigate these limitations, the authors propose a novel approach that generalises over the more widely-used compositional DGPs via a hierarchical model that incorporates locally linear deformations of stationary kernels. A synthetic example is crafted in order to showcase how TDGP improves over competing methods for problems exhibiting heavy non-stationarity. This is further complemented by an analysis on a real-world dataset, as well as 4 benchmark datasets from the UCI repository.

**Strengths:**

- The paper is very well-written and a pleasure to read. I commend the authors for structuring the paper in a manner that clearly shows connections to related models, while also showcasing the key contributions of this work.
- I particularly appreciated Sections 4.1 and 4.2, which capture the various dimensions along which TDGP can lead to improvements over other methods (predictive performance, uncertainty quantification, and interpretability).
- The problem statement tackled in this paper is well-motivated, and the limitations of other techniques are clearly pointed out both verbally and visually in the experiments section. One possible improvement to this could be highlighting application domains where TDGP is expected to be most impactful – i.e., which application domains tend to non-stationary data where well-calibrated uncertainty is especially useful. While fairly minor, this could help clarify the significance of this work earlier on the paper.

**Weaknesses:**

- The authors comment in the *Limitations* section that the increase in the number of variational parameters could slow down optimisation – it could be worth supporting this statement by specifying the computational complexity, and maybe an analysis reporting wall clock time for one of the experiments in Section 4.
- I would also be interested in whether the increased parametrisation of the model results in models that are less stable or difficult to consistently converge. This has implications on the practical utility of the model.
- The paper focuses heavily on using the SE kernel throughout – can this be extended to other kernels or is the value-add of this extension not considered to be worthwhile for the paper?

**Questions:**

I do not have additional questions beyond the points raised in the *Weaknesses* section. I would appreciate if the authors could read through this and indicate whether they can improve upon these aspects in a future revision.

The paper is a solid addition to the literature on hierarchical GP models – besides proposing a novel method, it also gather and illustrates potential deficiencies in existing methods that are widely-used in the community. I also believe that the paper is already in a state that is fit for publication, and can only be improved further with additional minor iterations.

---

> ### Author Rebuttal · Authors · 2023-08-09
>
> Thank you for your detailed feedback. We have run additional experiments to answer your questions (below). Besides including the additional results and complexity analysis to the final manuscript, we will also add a discussion on which domains we expect TDGP to be most useful, as suggested.
>
> >The authors comment in the Limitations section that the increase in the number of variational parameters could slow down optimisation – it could be worth supporting this statement by specifying the computational complexity, and maybe an analysis reporting wall clock time for one of the experiments in Section 4.
>
> Thanks for the suggestion. For simplicity, assume the number of latent dimensions is constant across layers. Regarding memory complexity, TDGP uses $L\times D\times Q\times m$ variational parameters for the mean and $L\times D\times Q\times m^2$ for the covariance. As for the computational complexity one ELBO evaluation costs $O(L\times D\times Q\times m^3)$ for the KL terms and $O(L\times D\times Q\times m^2\times n)$ for the expectation term. Although we did not link it properly in the main text, we have a wall-clock time analysis for all experiments in Appendix C, Tables 3 and 4. We will include the complexity analysis to the final manuscript. We will also add a reference to the wall-clock tables in the experiments section.
>
> >I would also be interested in whether the increased parametrisation of the model results in models that are less stable or difficult to consistently converge. This has implications on the practical utility of the model.
>
> As discussed in our limitations, we expect models with a higher number of parameters to be more difficult to optimize. However, Figure 1 in the rebuttal shows that TDGP is reasonably stable for moderately-sized values of $Q$. Since you have also asked for wall-clock times (above), we also included a complimentary plot with time as a function of $Q$ in Figure 1. As TDGP has an architecture that is not comparable to previous DGP models, best practices for learning rate scheduling and pre-training for increased robustness are still unknown and we intend to explore this topic in the future.
>
> >The paper focuses heavily on using the SE kernel throughout – can this be extended to other kernels or is the value-add of this extension not considered to be worthwhile for the paper?
>
> The inference scheme that we used in our experiments and as derived in Appendix B.3 is only valid for SE kernels due to the necessity of the computation of the $\Psi$-statistics, which are convolutions of the kernel with Gaussian densities. However, using solely the SE kernel is commonplace in the Deep GP literature, as the presence of the hidden layer alleviates the drawbacks of such a simple kernel. For example, the experiments done by Salimbeni & Deisenroth (2017) only consider SE kernels.
>
> Nonetheless, we can train our model using the doubly stochastic VI (DS-VI) scheme of Salimbeni & Deisenroth (2017), which allows us to benchmark different kernel choices. Note, however, that DS-VI is stochastic and black-box. Therefore, compared to our ELBO, DS-VI results in a loose lower-bound on the evidence. For completeness, we have run TDGP using DS-VI on the benchmark datasets for the SE kernel and the Matérn 3/2 kernel (Rebuttal, Figure 2). As expected, results were worse than with our tailored inference scheme.

---

> > ### Comment · Reviewer_DvNb · 2023-08-18
> > **Acknowledgement of rebuttal.**
> >
> > Thank you for your detailed replies - I have read the other reviews as well as the consequent discussions, and I still consider this paper to be a valuable addition to the literature on (deep) GPs. Coupled with the fact that the paper was already well-written and presented in its original submission, for the time being I am keeping my score for the paper.

---

### Official Review · Reviewer_HPwG · 2023-07-03

**Soundness:** 3 good
**Presentation:** 2 fair
**Contribution:** 3 good
**Rating:** 4
**Confidence:** 4

**Summary:**

This paper extends the CDGP to address two weaknesses in deep Gaussian processes. The solution is similar intuitively to a residual connection. Instead of letting each layer depend only on the last layer, they let each layer depend on the last layer and the input. This allows the model to induce a manifold and use a lengthscale field.

**Strengths:**

This seems like an important problem, and the solution seems principled.

**Weaknesses:**

There should be more discussion of what it means to not induce a manifold, why violating the triangle inequality leads to that, and why not inducing a manifold is a drawback. I think inducing a manifold implies that one can make the latent space plots: why are these valuable? Why do they require the triangle inequality?

There should be more discussion of how a lengthscale field leads to interpretability.

In both the latent space and the inverse lengthscale plots, there should be some discussion about why we want to see these plots: for instance, in figure 7, we have the two sets of plots: what can we glean from them about the data/problem/etc? Figure 8 has a little bit (the zone of high correlation extends into the mountains), but this is not obvious for a non-expert application reader. You could have a short tutorial perhaps of how to read these plots in the supplement.

**Questions:**

Most of my questions are in the weaknesses.

**Limitations:**

They have a good discussion of limitations: mainly that it is computationally expensive and leads to requiring more parameters.

---

> ### Author Rebuttal · Authors · 2023-08-09
>
> Thank you for your suggestions to improve our manuscript. We agree that elaborating further on the interpretation of the experiments can make it more accessible to the general public. In the next paragraphs, we elaborate further on the importance of both types of representation (manifold/latent space and lengthscale fields). While we cannot update the manuscript at this point, we will include this discussion in the camera-ready version along with a guide on how to read the plots.  We hope our answers have sufficiently addressed your concerns.
>
> >Manifolds, latent space plots and the triangle inequality
>
> Indeed, inducing a manifold implies there is some function $\tau$ mapping inputs to a finite-dimensional latent space. We can use the plots to analyze how $\tau$ changes the notion of neighborhood (or similarity) between points in the input space. For instance, Figure 7 reveals that TDGP learns to differentiate the mountains from both the landmass above and the water below, mapping the mountains to a different range of values. On the other hand, Figure 7 also shows that DKL and CDGP do not discriminate between the mountain range and the landmass. In general, we can use these plots to analyze $\tau$ and help human experts validate/understand/debug our models. It is not clear how this could be done for models without an explicit $\tau$, like DNSGP. In the case of DNSGP, the quadratic term in our Equation 3 ($\delta$) is what makes it hard to write the kernel as $k(\tau(a), \tau(b))$. To rewrite $\delta$ as a quadratic form in $\tau(a)$ and $\tau(b)$, we would need to redefine $\tau$ to take $a$ and $b$ at the as inputs at the same time, as discussed by Paciorek [19, page 32]. If $\delta$ induced a metric, writing DNSGP in terms of some $\tau$ would be simpler. Importantly, while it might be possible to extract some $\tau$ from DNSGP, how it is possible or how to do so is an open problem, as far as we are aware.
>
> >Importance of lengthscale fields for interpretability
>
> The inverse lengthscales allow interpreting the prior smoothness of a function at each input point. For instance, areas in which $W(x) W(x)^\intercal$ is “high” (sum of eigenvalues)  correspond to areas where the function should be less smooth, i.e. have increased variance in their derivatives. More concretely, the lengthscale plots in Figure 7 shows that TDGP captures the non-smoothness in the transitions between the landmass, the mountain and the water. A similar phenomenon happens for DNSGP. This is in contrast with the latent space plots which don't directly relate to properties of the learned function.
>
> We will add an extended description of how to interpret the lengthscale field to our appendix, as this might not be widespread knowledge.

---

### Official Review · Reviewer_CrCc · 2023-07-04

**Soundness:** 3 good
**Presentation:** 3 good
**Contribution:** 3 good
**Rating:** 4
**Confidence:** 4

**Summary:**

This paper considers the problem of non-stationary kernel design for Gaussian processes. Two main approaches for this problem are deformation kernels and length-scale mixture kernels. On one hand, deformation kernels, eg deep GPs, have found great success while trading off expressivity and the ability to learn latent manifolds in intermediate layers, for lack of interpretability and being prone to pathologies. On the other, length-scale mixture kernels retain interpretability but typically break the triangle inequality in the induced latent space, precluding a manifold structure. The authors propose thin and deep GPs, which aim to combine the benefits of both these approaches, while mitigating their drawbacks. The approach learns a position-dependent matrix, parameterized by a GP, which learns a pointwise linear transformation of the data. The output of such a layer can then be composed to learn a new matrix acting on the original data, enabling a deep learning structure. Numerical examples are provided which appear competitive with existing techniques.

**Strengths:**

This paper is clearly presented and combines existing ideas in a novel way. The numerical examples seem well selected, with a mixture of synthetic/analytic examples that demonstrate the utility of the method, with standard benchmark datasets allowing for meaningful comparisons on real data. A case study is also provided that is geared towards the strengths of the technique.

**Weaknesses:**

The main weakness of the paper in my view is Theorem 3.1. It appears that the proof that the CDGP is a generalization of deep GPs hinges upon the introduction of a new variable that plays the role of the deep GP, and then turning off their method's parameters by concentrating their priors around zero. This seems highly misleading unless the method is introduced in a way that uses the augmented space as default, or some numerical evidence is given that it should be turned off if appropriate. In this case, presumably the deep GP should also be tested with and without an augmented space for fair comparison. It seems that the augmented space is not used in any of the experiments in any of the models presented, since the local linearity leaving the data unchanged around zero is listed as a limitation.

**Questions:**

Have you tested the method with and without the data being augmented with a bias term? Has this been tested in prior methods as well?

Since the matrices are position dependent, the dimension of their range can vary across the domain as a different number of their singular values. How does this influence the latent manifold interpretation that motivated your work?

**Limitations:**

Besides those listed above, the computational expense is listed as a major limitation. This is a common drawback of related work, and it may be helpful to list the memory and computational complexity of this method with L layers, as is commonly done in the support literature.

---

> ### Author Rebuttal · Authors · 2023-08-09
>
> Thank you for the comments.
>
> We would like to highlight that the bias term is a simple design choice (used profusely throughout ML) that suffices to show TDGP is at least as expressive as CDGP. Both CDGP and TDGP with no bias are limiting cases of TDGP with bias.
>
> > Have you tested the method with and without the data being augmented with a bias term? Has this been tested in prior methods as well?
>
> Empirical performance may often rely on factors that are alien to theoretical analysis — e.g., inference quality, initialization, learning rate scheduling. In particular, adding bias terms increases the number of parameters of each layer $\ell$ in $Q_\ell$, which makes inference more challenging and is the main reason we did not test it before the submission. We have run TDGP with bias on the synthetic experiment using doubly-stochastic inference. As expected, results were significantly worse when we included the bias (NLPD: 0.7 vs -0.3). However, if we reduce the number of parameters  restricting $W(\cdot)$ to be diagonal, TDGP w/ bias outperforms TDGP w/o bias (NLPD: -2.7 vs 1.8). This corroborates our hypothesis.
>
> > Since the matrices are position dependent, the dimension of their range can vary across the domain as a different number of their singular values. How does this influence the latent manifold interpretation that motivated your work?
>
> Formally, to characterize a manifold, we need the number of singular values of $W(x)$ to be a constant for all $x$ in the domain. The restriction that the local dimensionality must be constant is usually not  enforced in most deep learning models. In particular, this also applies to CDGP. However, even if this quantity fluctuates, it is still reasonable and commonplace to use the latent embedding plots to interpret the model. Nonetheless, unlike CDGP, our model provides a straightforward way to quantify the local dimensionality. Therefore, we also include Figure 3 with the distribution of non-zero singular values in the data for different folds and datasets and also Figure 4 with the average singular values in the data, showing their correlation with Figure 10 in the paper.
>
> >This is a common drawback of related work, and it may be helpful to list the memory and computational complexity of this method with L layers, as is commonly done in the support literature
>
> Thanks for the suggestion. We will include the spatial and temporal complexity analysis to the final manuscript, as they are only implied by the description of the model. Just as shown in Figure 1 of the rebuttal, we expect increases in depth to linearly increase the time spent in the optimization of the ELBO.

---

> > ### Comment · Reviewer_CrCc · 2023-08-18
> > **Response**
> >
> > I have read your response as well as the responses to the other reviewers. I would suggest the authors take a closer look at their construction/justification as a generalization of CDGPs before revising this work, and consider changing the way this is pitched. I would like to make it clear that I think the idea is a good one, but the presentation needs work.

---

> > > ### Author Response · Authors · 2023-08-18
> > >
> > > We are glad you think our paper's idea is good. We would be grateful for further details/suggestions on how to improve the specific issues you noticed and how you would suggest changing our pitch, as we want to address your concerns and present our ideas in the best possible way.

---

> > > > ### Comment · Reviewer_CrCc · 2023-08-19
> > > > **Theorem 3.1**
> > > >
> > > > My main apprehension is with Theorem 3.1 as I stated in the initial review. The justification as a generalization of CDGPs seems flimsy at best. While it is true that bias terms exist elsewhere in machine learning, introducing an artificial term that allows you to introduce the other model, then turning your model off, and then claiming that this makes CDGPs a special case of TDGPs seems dishonest. It would be a different story if the intermediate models performed well, even if the best performance is at the endpoints. However if that were the case you would be writing about the whole family of models rather than the model with the `bias' term removed. This is reflected in the experiments you mentioned in your response.
> > > >
> > > > In my view the inclusion of this justification weakens the entire paper. It seems too forced, when no such theoretical justification as a generalization of CDGPs is required in order for your model to be a valid contribution, and it is not an important fact. While your construction is more refined that interpolating between your model and an arbitrary one, then setting parameters to turn the other one off, it has the same flavor and cheapens your otherwise good work.

---

> > > > > ### Author Response · Authors · 2023-08-20
> > > > > **Recontextualizing Theorem 3.1 for the next revision of the main text**
> > > > >
> > > > > Thank you for detailing your concerns. We want to highlight that this claim is only present in select sections, as part of the contribution section, the theorem statement, and the conclusion. Nonetheless, our goal with this theorem was to prove that our proposed model (TDGP without bias) has a deep connection with the usual CDGP model, as both can be considered different limits of a more general model (TDGP with bias). This property is not shared with other non-stationary models, such as DNSGP, and we believe that this perspective of the deep models as locally affine maps is of interest to the DGP community.
> > > > >
> > > > > Regardless, we still object to the impression that TDGPs as a generalization of CDGPs is dishonest, as the introduction of the $d(\cdot)$ terms is not artificial or forced by us but a consequence of the well-known connection between linear transformations (TDGP) in $R^d$ and affine transformations (CDGP) in the projective space of $R^{d+1}$. We emphasize that the only modeling choice in the theorem is adding bias and setting the last row of $\tilde{W}$ to $[0, 0, \ldots, 1]$ (otherwise the bias dimension would not be 1 in the next layer) and that these choices are valid under the description of the TDGP model in equation 8, for example.
> > > > >
> > > > > However, in the interest of dispelling possible misunderstandings, we would like to rephrase the lines where this theorem is referenced:
> > > > >
> > > > > * (Lines 53 & 54) **We prove that TDGPs and compositional DGPs are the limits of a more general DGP construction. Thus, we establish a new perspective on standard CDGPs while reaping the benefits of inducing a lengthscale field.**
> > > > > * (Line 130) **Importantly, however, TDGP are related to CDGPs as both can be seen as locally affine deformations of the input space (proof in Appendix A)**
> > > > > * (Lines 258-260) **Since TDGP is based on deformation kernels and is related to CDGPs, we believe it can leverage many of these advances in inference, which is left to future investigations.**
> > > > > * (Lines 272-274) **Additionally, while TDGP shares a connection with CDGP, our prior does not concentrate on "flat" samples when the number of layers increases — as is the case with CDGP**
> > > > >
> > > > > Would these changes ease your concerns and elevate the overall contents of the paper?

---

### Official Review · Reviewer_6DJk · 2023-07-13

**Soundness:** 3 good
**Presentation:** 4 excellent
**Contribution:** 3 good
**Rating:** 7
**Confidence:** 3

**Summary:**

The paper presents a new deep Gaussian process (DGP) model, the thin and deep GP (TDGP), which does not suffer from a diminishing signal as the number of layers increases. The crux of the TDGP model is the covariance function that, for each layer $\ell$, acts on a linear combination of a (non-linear) transformation of the outputs of the previous layer and the inputs $\mathbf{x}$, $\mathbf{h}^\ell(\mathbf{x}) = \mathbf{W}^\ell(\mathbf{h}^{\ell-1}(\mathbf{x})) \mathbf{x}$. This hierarchical construction is shown to cover the usual DGP models while not suffering from diminishing signals as the depth increases. To perform inference in this model, the authors adopt a mean-field VI scheme, which is demonstrated to be effective through a number of experiments on both synthetic and real-world data.

**Strengths:**

1. The paper presents an interesting and original approach to DGPs, an important field that should interest the NeurIPS community.
2. The paper is of high overall quality and very well-written.
3. The presented model has the potential to be of considerable significance, given that it generalises previous DGP models while not suffering from the same pathologies.

**Weaknesses:**

1. While I think the paper presents a great idea with lots of potential, my main concern is the lack of a deeper analysis of the model. Given that the proposed model is essentially just a particular transformation of the inputs at each layer and a rather straightforward mean-field VI scheme, I would expect some insights into the behaviour of the model, even if these were just empirical. The current experiments do present some interesting insights, but, say, what happens to the model's performance as we increase the depth of the model? What if we change the width of the layers? What if we use different covariance functions than the squared exponential? How does the induced covariance matrix look at different layers of the model compared to, say, DNSGP? Why does the model encourage low-dimensional latent spaces, and does this happen too for models deeper than just two layers? A Gaussian prior shouldn't encourage sparsity, so this aspect of the model is particularly puzzling.
2. As mentioned, a standard mean-field VI scheme seems very rough given that this is known to perform poorly for many models, including DGPs. To my knowledge, the current state-of-the-art is still the doubly stochastic VI scheme by Salimbeni & Deisenroth (2017), so it would make sense to try something similar (and unless I misunderstand something, it seems fairly straightforward to do this for the proposed model).
3. One clarity issue (perhaps the only one) I find with the paper is that it is unclear to me which DGP model the authors refer to as "CDGP". Deep GPs have evolved dramatically from the original formulation of Damianou & Lawrence (2013), and the mentioned pathologies are not as pronounced anymore (to my knowledge). It is great that the authors compare against DNSGP, but comparing against the doubly stochastic DGP rather than the original formulation makes much more sense. As it is unclear if the authors actually compare to the current state-of-the-art, it is also unclear if the proposed model addresses actual problems.
4. Experimentally, the model seems to work well, but the authors use only two layers for all models. This is particularly strange as a key selling point of the proposed model is that it doesn't degenerate as the number of layers increases. But it is also an odd choice since deeper models should work better (and DNSGP demonstrably does for two of the UCI datasets). It is also strange that MRAE statistics are only reported for the GEBCO dataset; these should also be reported for the UCI experiments.
5. The model is claimed to be "interpretable", but this only seems to be the case for a two-layer model (i.e., one with a single latent space). For deeper models, the distance matrix is still a highly non-linear function of the inputs, which I don't think will be particularly interpretable.


References:

- Andreas Damianou & Neil Lawrence, "Deep Gaussian processes", AISTATS 2013.
- Hugh Salimbeni & Marc Peter Deisenroth. "Doubly Stochastic Variational Inference for Deep Gaussian Processes", NeurIPS 2017.

**Questions:**

1. Which specific variation of the DGP model is referred to as "CDGP"?
2. In lines 138-139, you say that "By placing zero-mean priors on the entries of $\mathbf{W}$, we encode an inductive bias to maximally reduce the latent dimensionality of $\mathbf{h}$." Why is this? Intuitively, since the prior is Gaussian, it shouldn't encourage sparsity.
3. Your contribution seems to be an input transformation, which isn't tied to a specific covariance function, so is there a reason that you chose the squared exponential (or exponentiated quadratic) covariance function? The Matérn class is more often used in practice, I believe.
4. Is a model with more than two layers still interpretable? If so, how would you interpret the latent space?
5. In Figures 4 and 7, how is the inverse length scale computed for TDGP? Also, why can we not compute the inverse length scales for CDGP or DKL? And why can we not visualise the latent space for DNSGP?
6. In lines 51-52, you mention that TDGP is the only deep architecture that induces data embeddings. What do you mean by "data embedding" here, and why do other models not have this property? For instance, even for the original DGP model, we can easily visualise the distribution of a particular sample at a particular layer.

**Limitations:**

The authors have adequately addressed limitations in a dedicated paragraph.

---

> ### Author Rebuttal · Authors · 2023-08-09
>
> Thank you for the review. We are glad you found our paper to be interesting, significant, and of high quality. We appreciate your concerns and questions and hope you will find them addressed in the following:
> ## Approximate inference and kernel choice (W2, Q3)
> We derived our ELBO in Appendix B.3 for the two-layer model. This gives a collapsed bound which is superior to the doubly stochastic (DS-VI) scheme of Salimbeni & Deisenroth (2017) in small to medium scale regression settings, but relies on Gaussian likelihood and squared exponential kernels. However, we can train our model using the DS-VI scheme for arbitrary likelihoods, kernel functions, and depths.
>
> As seen in Figure 2 of the rebuttal, we ran experiments with DS-VI for the TDGP using both squared exponential and Matérn-3/2 kernels; the results are worse than our original results due to the stochasticity in the DS-VI ELBO.
> ## Induced covariance matrix (W1) and deeper models (W4)
> We include a comparison of the induced covariances at different depths in the rebuttal's Figure 5. CDGP's covariances eventually "flatten" out due to contraction of the image, and the prior covariance gets stuck. In contrast, DNSGP and TDGP covariance can never get stuck: even if a previous layer becomes constant, this is not an issue, as each layer is always a function of the input, as the range of one layer does not depend on the previous.
>
> Thus, TDGP with increasing depth does not degenerate, as opposed to CDGP. Nevertheless, in the current work we focused on deriving and experimenting with the collapsed ELBO for a 2-layer TDGP. We emphasize that a similar ELBO can still be obtained for more layers if a SE kernel and a Gaussian likelihood is considered. Further experiments with deeper models are left for future investigations due to their complexity.
> ## Clarification on CDGP (W3, Q1)
> By CDGP we refer to the general prior model with stacked GP layers; this describes the bulk of DGP works since Damianou & Lawrence (2013). Most of the advances in DGPs have been in improving approximate posterior and inference for this model class. In our experimental comparison, we indeed use the doubly stochastic DGP with a linear mean function.
>
> We will clarify the methods used for approximate inference in each model.
> ## Other metrics (W4)
> As for the metrics, we did not report MRAE statistics for the UCI experiments due to layout constraints and we believed NLPD would be the most relevant metric to show. However, we will add a table in the appendix with the MRAE scores.
> ## Why TDGP induces sparsity (W1, Q2)
> Thanks for bringing this to our attention, we acknowledge that lines 136-141 are not as clear as we would like. Indeed, a Gaussian prior does not encourage sparsity. However, every entry of the $W(x)$ matrix is an independent Gaussian process with its own kernel function (Eq. 8). Each kernel has a hyperparameter $\sigma_{w_ij}^2$ which controls the prior variance of $w_{ij}(x)$. As usually in GP models, we optimize these hyperparameters with respect to the ELBO.
>
> If $\sigma_{w_ij}^2$ tends to zero, then the posterior of $w_{ij}(x)$ concentrates on the prior mean. If, as discussed in lines 136-141, we set the prior mean to zero, then $\sigma_{w_ij}^2\to 0$ corresponds to those entries of the $W(x)\to 0$, so a whole row tending to zero can effectively remove a dimension of the latent space. This effect is explained in lines 199-202 and it can be visualized in practice as shown in Figures 5 and 10.
>
> We will clarify this in the final version.
> ## Interpretability in deeper models (W5, Q4)
> As argued in lines 104-119, $h^\ell$ being a locally linear transformation means we can analyze each $W^\ell$ as a function of the original input space $x$ with lengthscale field $W^{\ell-1}(x)$. Take a TDGP model with three layers, $f(W^2(W^1(x) \cdot x) \cdot x)$, as an example. When analyzing the last layer $f$, we can reproduce the same analysis of Figures 4, 7 and 8 and plot the output of the hidden layer $W^2(x)$ to analyze the lengthscale field of $f$: areas of high inverse lengthscale correspond to areas with high variance for the derivative of $f$ and vice versa. The same analysis can be repeated for $W^2$ and _its_ lengthscale field defined through $W^1$. This analysis holds since for TDGP each hidden layer is a locally stationary function with respect to the original input space.
>
> We will add an extended description of how to interpret the lengthscale field to our appendix.
> ## Computation of lengthscale field (Q5)
> As mentioned in line 113, $[W(x)^\intercal W(x)]^{-1}$ is the lengthscale field, and correspondingly the local inverse lengthscale is computed as $\Phi(x) = W(x)^\intercal W(x)$. In Figures 4 and 7, we plot the trace of this $\Phi(x)$ for the last layer, which reflects the smoothness of the output around each point. We have clarified this in the text.
>
> This cannot be computed for CDGP or DKL, as both use deformation kernels of the form $k(\tau(a), \tau(b))$, which cannot in general be represented as a function of $(a - b)\Phi(a,b)(a - b)$ nor of $\| \Phi(a)^½ a - \Phi(b)^½ b \|$ for some appropriate inverse lengthscale function $\Phi$.
>
> ## Latent space (Q5, Q6)
>
> > why can we not visualise the latent space for DNSGP?
>
> If we can write the kernel as $k(a,b) = k’ (\tau(a), \tau(b))$, then the deformation $\tau(x)$ defines the latent space. However, the $\delta$ term at the core of DNSGP (see our Eq. 3 and line 86) makes this impossible.
>
> > In lines 51-52, you mention that TDGP is the only deep architecture that induces data embeddings.
>
> We meant that TDGP is the only architecture that induces data embeddings **and** lengthscale field. We have clarified in the text that this means _at the same time_. By “data embedding” we mean that the models with successive transformations of the original data through the layers of the model, as in the DKL and CDGP models (but not in DNSGP).

---

> > ### Comment · Reviewer_6DJk · 2023-08-18
> >
> > Dear authors,
> >
> > Thank you very much for your elaborate answers and clarifications, and many apologies for not getting back to you sooner.
> >
> > After reading your (impressive!) rebuttals to myself and the other reviewers, I feel much more positive about the paper. You have clarified many things for me and addressed my concerns in an impressive way. In particular, if you can add discussions like the one you have provided in your response to my questions regarding the induced covariance matrix (W1) and deeper models (W4), I think your paper will be stronger. I understand that you might want to keep the theoretical analysis to just two layers, but showing experiments with more, as you have done in your rebuttal, would at least reassure the reader that the model is not limited (in practice) to two layers. Having these in the supplementary would be just fine.
> >
> > The additional analyses and explanations you have provided to all of us here have given me a significantly better grasp of the paper, and I encourage you to incorporate them into your paper (or appendix). I have updated my score and will recommend acceptance.
> >
> > A minor thing I have just noticed, in line 172, you refer the reader to the appendix for additional information regarding the synthetic experiment, but I can't seem to find it there.

---

> > > ### Author Response · Authors · 2023-08-18
> > >
> > > Thank you for recommending the acceptance of our paper and alerting us about the missing details concerning the synthetic experiment in the appendix; we will correct this issue and reference the additional figures in the main text and the supplementary material.

---

### Author Rebuttal · Authors · 2023-08-09

We thank the reviewers for their thoughtful feedback, as we believe addressing them will broaden the audience for this paper. We are also glad that most reviewers feel our ideas are clearly presented and constitute a solid and principled contribution to this problem space.

In particular, the considerations shared across reviewers which will be revised in the final manuscript are:
* further explanation on the interpretability of our model and lengthscale fields in general,
* inclusion of big-O notation complexity analysis,
* further explanation on the possible use of alternative kernels and inference methods.

As for additional plots and experiments, our attached PDF includes experimental analysis relating to:
* (Figure 1) convergence and wall clock timing as a function of TDGP’s width,
* (Figure 2) additional experiments with alternative inference schemes and non-SE kernels,
* (Figures 3 and 4) an analysis of the local dimensionality of TDGP’s latent space,
* (Figure 5) an illustration of the behavior of the prior covariance for different numbers of hidden layers.

In each review’s personal rebuttal, we address in detail each question and concern.

---

### Decision · Program_Chairs · 2023-09-21

**Decision:**

Accept (poster)

**Comment:**

New approaches to non-stationary kernel design and (variational) inference are presented. Contributions and limitations are described and it is well written. However, the rebuttal process has lead to better explanations/presentation suggestions; I strongly suggest updating the final version that incorporates it carefully.